# Notch Signaling Regulation in Autoinflammatory Diseases

**DOI:** 10.3390/ijms21228847

**Published:** 2020-11-23

**Authors:** Rossella Gratton, Paola Maura Tricarico, Adamo Pio d’Adamo, Anna Monica Bianco, Ronald Moura, Almerinda Agrelli, Lucas Brandão, Luisa Zupin, Sergio Crovella

**Affiliations:** 1Institute for Maternal and Child Health—IRCCS “Burlo Garofolo”, 34137 Trieste, Italy; rossella.gratton@gmail.com (R.G.); adamopio.dadamo@burlo.trieste.it (A.P.d.); annamonicarosaria.bianco@burlo.trieste.it (A.M.B.); ronaldmoura1989@gmail.com (R.M.); lucabrand@gmail.com (L.B.); luisa.zupin@burlo.trieste.it (L.Z.); 2Department of Medical Surgical and Health Sciences, University of Trieste, 34149 Trieste, Italy; 3Department of Pathology, Federal University of Pernambuco, Recife 50670-901, Brazil; almerindaapimentel@gmail.com; 4Department of Biological and Environmental Sciences, College of Arts and Sciences, University of Qatar, Doha 2713, Qatar; sgrovella@qu.edu.qa

**Keywords:** Notch pathway, autoinflammation, genetic, autoinflammatory diseases

## Abstract

Notch pathway is a highly conserved intracellular signaling route that modulates a vast variety of cellular processes including proliferation, differentiation, migration, cell fate and death. Recently, the presence of a strict crosstalk between Notch signaling and inflammation has been described, although the precise molecular mechanisms underlying this interplay have not yet been fully unravelled. Disruptions in Notch cascade, due both to direct mutations and/or to an altered regulation in the core components of Notch signaling, might lead to hypo- or hyperactivation of Notch target genes and signaling molecules, ultimately contributing to the onset of autoinflammatory diseases. To date, alterations in Notch signaling have been reported as associated with three autoinflammatory disorders, therefore, suggesting a possible role of Notch in the pathogenesis of the following diseases: hidradenitis suppurativa (HS), Behçet disease (BD), and giant cell arteritis (GCA). In this review, we aim at better characterizing the interplay between Notch and autoinflammatory diseases, trying to identify the role of this signaling route in the context of these disorders.

## 1. Introduction

Notch signaling is an evolutionarily conserved pathway exerting a pivotal role in mediating the regulation of a vast variety of cellular functions and responses, both during normal development and pathological conditions, through a relatively simple signal transduction route in which few components act synergically with the aim of transmitting the signal from the cell surface to the transcriptional machinery [1]. Upon activation, Notch pathway regulates fundamental cellular processes such as differentiation, migration, proliferation, cell fate and death, which have been thoroughly described in the past years [1]. Nevertheless, growing evidences indicate the presence of a strict crosstalk and association between Notch signaling and inflammatory responses, that seems to be mainly driven in immune cells by the direct functional role of Notch on the modulation of cellular functions (proliferation, differentiation, cell fate, and apoptosis) involved in inflammatory processes [2].

Autoinflammatory diseases (AIDs) are a large family of heterogeneous disorders characterized by periodic episodes of spontaneous and aberrant sterile inflammation [3].

Considering the role of the Notch pathway during inflammatory responses and the continuously expanding spectrum of AIDs, it is important to deepen the knowledge related to the mutual interactions between AIDs and Notch signaling.

In order to address this topic and to depict an overview of the current data on this issue, first, we queried the NCBI PubMed database by employing the keywords “autoinflammatory disease AND notch signaling” (https://pubmed.ncbi.nlm.nih.gov/?term=autoinflammatory+disease+and+notch+signaling&sort=date). Remarkably only three articles regarding three distinct disorders were retrieved and include: hidradenitis suppurativa (HS); Behçet disease (BD); medium and large vessel vasculitis, namely giant cell arteritis (GCA). Therefore, in this review we aim at analysing and illustrating the pathogenesis of these three AIDs, focusing on the direct involvement of the Notch pathway in order to attempt a deeper understanding of the role of this signaling route in the context of these disorders. 

### 1.1. Autoinflammatory Diseases

Autoinflammatory diseases (AIDs) encompass a family of heterogeneous conditions caused by a defect or a dysregulation of the innate immune system. AIDs are characterized by recurrent episodes of aberrant and spontaneous sterile inflammation involving multiple organ systems, and by the lack of high autoantibody titers or antigen-specific T lymphocytes [3].

To date, the rapidly growing knowledge in the field of innate immunity ensures that the classification of AIDs is a spectrum in a state of continuous expansion that possesses a progressively growing level of complexity [4,5]. In this context it is important to understand the difference between autoinflammation and autoimmunity. Autoinflammation is defined as a dysregulation of innate immunity and inflammation in which the prefix “auto” specifies the automatic/autonomous origin of the response. Autoimmunity is characterized by a dysregulation of adaptive immunity and the prefix “auto” identifies the self-origin of the immune response [6,7].

AIDs are characterized by injuries in multiple organ systems that often lead to tissues damage, i.e., danger signals such as nonmicrobial damage-associated molecular patterns (DAMPs) or microbial pathogen-associated molecular patterns (PAMPs), are recognized by pattern recognition receptors (PRRs), comprehending toll-like receptors (TLRs) and/or nucleotide-binding oligomerization domain-containing (NOD)-like receptors (NLRs). Activation of these receptors induce the release of proinflammatory cytokines and occurs mainly in innate immune cells in particular in neutrophils, monocytes, macrophages, as well as in mast cells [8].

AIDs are also considered to be interleukin (IL)-1-mediated diseases that respond to anti-IL-1 therapies; in fact, autoinflammation is characterized by an overactivation of the innate immune system and by the overproduction of proinflammatory cytokines including IL-1β, IL-18, tumor necrosis factor (TNF), and type-1 interferons (IFNs) [9,10,11]. IL-1β and IL-18 can be released by the inflammasome NLR family pyrin domain containing 3 (NLRP3), an intracellular molecular sensor that promotes inflammation and also programmed cell death [12,13]. Another important pathway in the autoinflammatory process is the nuclear factor-kB (NF-kB) route. The NF-kB pathway is regulated through multiple post-translational mechanisms such as ubiquitination, and it plays an important role in triggering the inflammatory process by inducing the expression of proinflammatory chemokines and cytokines comprehending IL-1, IL-6, TNF and also by participating in inflammasome regulation [14,15,16,17,18].

Currently, AIDs include monogenic syndromes as well as polygenic and multifactorial diseases [19]. AIDs monogenic syndromes are hereditary diseases due to mutations in a single gene that are characterized by childhood onset, recurrent episodes of fever, skin manifestations, and disease-specific patterns of organ inflammation [20]. Multifactorial polygenic diseases comprise conditions possessing a clear autoinflammatory aetiology but with no specific monogenic identified cause, in which interactions between gene alterations, environment, and lifestyle contribute to the development of the disease. The disorders included in this group are characterized by nonspecific symptoms that comprise recurrent flares or persistent systemic inflammation, fever, skin manifestations, chest and abdominal pain, lymphadenopathy, and arthritis [21,22] (Figure 1).

Multifactorial polygenic AIDs, more common than the monogenic ones, have recently numerically expanded since an increasingly growing number of novel diseases with genetic or idiopathic origin have been included in this group of disorders. Moreover, the exact pathogenetic mechanisms at the basis of many multifactorial polygenic AIDs are still largely unknown. A further level of complexity is given by the presence of many different forms of multifactorial polygenic AIDs, diversity primarily given by the involvement of differential molecular mechanisms underlying the diseases’ onset and progression, as well as the possible intervention of environmental and behavioural factors that still need to be unravelled.

### 1.2. Notch Signaling

Notch signaling is an evolutionarily conserved intracellular cascade that regulates a wide variety of cellular processes [23]. Considering the well-established impact of Notch signaling in the maintenance of the homeostasis of various cellular functions, it is not surprising that variations, in terms of altered regulation or direct mutations in the core components of Notch signaling, may lead to an aberrant activity of the intracellular signaling route and to the development of several diseases [24].

In mammals, four different Notch receptors have been described (Notch 1–4) [25]. Structurally, all Notch receptors are single-pass type I integral membrane proteins consisting of an extracellular ligand-binding domain (NECD) (N-terminal), a single transmembrane domain (TMD), and an intracellular domain (NICD) (C-terminal) [26]. Dissimilarities amongst Notch1–4 receptors reside primarily in structural differences in the NICD, a region that is known to directly affect the binding affinity of the extracellular partition of the receptor with its ligand and the interactions of receptors with transcriptional factors in the nucleus upon their activation, thus modulating the expression of downstream target genes [27].

Notch signaling is triggered by five single-pass integral membrane ligands belonging to the Serrate family of ligands (Jagged1 and Jagged2) and to the Delta-like family of ligands (delta-like 1 (DLL1) ligand, delta-like 3 (DLL3) ligand, and delta-like 4 (DLL4) ligand), which collectively belong to the Delta/Serrate/Lag-2 (DSL) family of ligands and are generally referred to as DSLs [28]. 

The activation of Notch signaling depends on a direct cell-to-cell contact in which transmembrane receptors are activated by transmembrane ligands on neighbouring cells [29,30]. The binding of Notch receptor to a DSL ligand expressed on a juxtaposed cell, leads to a proteolytic processing of the receptor by metalloprotease a disintegrin and metalloprotease (ADAM)-type, which removes the extracellular domain bound to the DSL ligand and releases the intracellular active fragment, namely the NICD [26,31,32]. Once released, the NICD translocates to the nucleus and binds to the conserved DNA binding transcription factor RBP-Jκ (recombination signal binding protein for immunoglobulin kappa J region), which is required for both repression and activation of Notch target genes [33]. In the absence of a signal–receptor interaction, RBP-Jκ represses transcription by interacting with the co-repressor (Co-R) complex containing a histone deacetylase. The binding of NICD to RBP-Jκ in the nucleus converts RBP-Jκ from a repressor to an activator of transcription through the recruitment of histone acetyltransferases (HAc) and the activation of the Mastermind-like (MAML) family of transcriptional activators. The association between NICD, RBP-Jκ, and MAML generates the ternary Notch transcription complex (NTC), which is further involved in the recruitment of general transcription factors involved in the transcription of downstream primary target genes, such as hairy/enhancer of split (Hes) and the hairy/enhancer of split with YRPW motif (Hey) [34,35]. Hes and Hey proteins are transcription repressors that negatively regulate the expression of downstream target genes including tissue-specific transcription factors that have a critical role in various cellular and developmental decisions regarding heart development, neurogenesis, and blood vessel formation [36] (Figure 2).

## 2. Autoinflammatory Diseases with Notch Pathway Involvement

Although the precise molecular mechanisms underlying the interplay between Notch signaling and inflammation is not yet fully understood, currently, the most accredited hypothesis assumes that, in an inflammatory environment, Notch signaling might be stimulated both by endogenous factors such as cytokines and exogenous components including pathogens [37].

Considering the important role of the Notch pathway during inflammatory responses and the continuous growing complexity of the AIDs, scientific literature data have recently begun to intersect the Notch pathway with AIDs. Although their number is destined to be enlarged, to date, few autoinflammatory disorders characterized by alterations in Notch signaling pathway are known and include hidradenitis suppurativa, Behçet’s disease, and giant cell arteritis.

### 2.1. Hidradenitis Suppurativa (HS)

Hidradenitis suppurativa (HS) is an inflammatory chronic condition characterized by profound inflamed nodules, abscess and sinus tracts, that may ultimately result in fibrosis and scarring, mainly affecting the inguinal axillary and submammary areas of the skin [38]. Despite its relative high prevalence (5–400:100.000) in the European population [39], very often HS is not easily identified by clinicians, thus, leading to an alarming diagnostic delay of seven years [40].

Although, traditionally, HS is defined as a pathology affecting only the epidermal tissue, recent evidences suggest that HS should be considered to be a systemic autoinflammatory disease with comorbidities influencing its manifestations [41]. Indeed, currently, HS is included in the autoinflammatory keratinization disorders (AiKDs) [42], a subgroup of AIDs characterized by a significant interplay between various genetic factors causing autoinflammation mainly in the epidermis and in the most superficial layers of the dermis. The induced autoinflammatory responses ultimately result in an aberrantly upregulated keratinization, thus, leading to the additional inflammatory symptoms of AiKDs [43].

The etiology of HS is still not completely unravelled, nevertheless, it is widely accepted that a complex multifactorial pathogenesis is at the basis of HS development [44].The keratosis and hyperplasia of the hair follicles lead to follicular occlusion and to cyst development followed by the subsequent scattering of the cysts’ content, mainly cellular debris and keratin filaments, in the surrounding dermis. These events can activate an inflammatory and immune response [45] involving different signaling pathways, such as the T-helper 17/IL-23 axis, the NLRP3 inflammasomes platform, TLRs, as well as immune cells (monocytes, macrophages, neutrophils, lymphocyte B, plasma cells, lymphocyte T, and natural killer cells) [44]. To note, in HS wounds an increment of TNF-α, IL-1ß, IL-17, IFN-γ [46], human ß-defensin, and cathelicidin LL37 were detected [47].

Moreover, multiple evidences have also highlighted the potential association between alterations in Notch signaling and the aberrant inflammatory response observed in HS patients. Notably, about 35% of HS cases disclose a familiar history of the disease and the typical pool of mutations found in familial cases occurs primarily in the *NCSTN, PSENEN*, and *PSEN1* genes. These genes encode respectively for nicastrin, presenilin enhancer 2, and presenilin 1 that are crucial subunits of the *ϒ*-secretase, a multifunctional complex involved in the proteolytic cleavage of transmembrane proteins including Notch receptor, amyloid precursor protein, members of the cadherin family and TLRs [48,49]. In addition, some sporadic (5%) HS cases have been reported to carry mutations in these same genes [50]. Mutations in the *ϒ*-secretase complex have a damaging impact on Notch signaling, thus, causing the disease [51]. Interestingly, *Notch1–/Notch2–* murine models show a keratinization impairment and a dermatological phenotype similar to the one registered in human HS cases [52].

Moreover, mutations in *POFUT1* and *POGLUT1* genes, encoding, respectively, for GDP-fucose protein O-fucosyltransferase 1 protein and protein O-glucosyltransferase 1, two proteins associated with Notch signaling, have been described in patients suffering with HS and Dowling–Degos disease, an autosomic dominant inherited skin disease that can occur alone or in association with HS in which patients present flexural hyperpigmentation [53].

In HS lesional skin, an increment of IL-23 produced by macrophages was reported. This cytokine stimulates the differentiation of naive T cells in Th17 lymphocytes, the latter are responsible for driving an autoinflammatory reaction in HS [54]. It has also been suggested that an impaired Notch signaling might have an impact on the functionality of macrophages [55]. Indeed, under normal conditions, Notch inhibits TLR-activated macrophages and MAPK-dependent proinflammatory cytokines production, thus, suppressing the Th17-mediated immune response [55]. A decrement of IL-22 was also observed in lesional HS skin [56], and IL-22 secretion has been shown to be normally dependent on Notch signaling activation in CD4+ T cells [57]. Furthermore, the reduced natural killer (NK) cells in HS [58] could again derive from impaired Notch signaling, since Notch promotes the functionality NK cells [59].

To date, it is widely accepted that a deficient Notch signaling characterizes HS etiopathogenesis, unfortunately the knowledge of how this impairment occurs is only partially known. A major level of complexity is given by the necessity to unravel the eventual interactions between Notch impairment and other factors (i.e., inflammation, innate immunity response, habits such as smoking, diet, etc.) that can contribute to the onset, progression, and severity of the phenotype.

### 2.2. Behçet’s Disease

Behçet’s disease (BD) is as a multisystem polygenic autoinflammatory disorder of unknown aetiology primarily characterized by the presence of periodic episodes of genital ulcers, oral aphthous ulcers, and uveitis, and therefore has been originally defined as a triple symptom disease [60,61]. Nevertheless, BD might also affect potentially all body organs and skin lesions, renal, cardiac, vascular, gastrointestinal, urologic, pulmonary, neurological, and arthritic manifestations have also been described [60]. Despite possessing a worldwide distribution, BD is commonly defined as Silk Road diseases, due to its high recurrence in the Mediterranean countries, Far East Asia, and in the Middle East, geographic areas that are traditionally considered as endemic for the disorder, in which the prevalence is estimated to range between 8 and 370/100,000 persons [61,62,63]. The onset of BD seems to affect females and males equally and usually occurs between the second and fourth decade of life [61]. The diagnosis of BD is critical due to the heterogeneous initial clinical manifestations, to the lack of pathognomonic tests, and to the presence of many possible differential diagnoses. Not surprisingly, on average, BD cases face a delay of several years in the establishment of an ultimate diagnosis from the initial onset of symptoms [61,64].

Though the pathogenesis of BD is yet not fully characterized, several studies have highlighted that, in genetically predisposed individuals, the presence of a complex interplay between the genetic background and a disrupted and augmented immune response that might be triggered by environmental factors such as various infectious-associated components, both auto-antigens and antigens, might induce the recurrent and abnormally increased inflammatory attacks typically linked to the onset and progression of BD [65].

Interestingly, BD stands on the border between autoimmune and autoinflammatory disorders. Indeed, BD is known to possesses responses that are characteristic of autoimmunity including the following: tight association with the *HLA-B*51* allele of the major histocompatibility complex, defined as the most distinctive genetic marker linked to BD and responsible for the typical phenotype of the disorder [65,66]; direct involvement and activation of Th1 lymphocytes, originally considered as the main cellular lineage involved in the diseases’ onset, and high levels of associated cytokines such as IL-18, INF-ϒ, IL-2 and IL-12 [65,67]; and hyperactivation of Th17 lymphocytes with consequent elevated production levels of IL-21, IL-2, IL-23, IL-17F and IL-17A, IL-6, TNF, and transforming growth factor-β (TGF-β) [68,69].

Nonetheless, recent findings suggest that BD also retains autoinflammatory properties such as the following: neutrophilic intrinsic hyperactivation which is presumably associated with incremented levels of reactive oxygen species production, phagocytosis, and chemotaxis; periodic events of remission and aggravation; absence of autoantibodies or autoimmune T and B cells [70]; and occurrence of recurrent apparently unprovoked self-limited amplified inflammatory responses to nonspecific stimuli characterized by augmented levels of proinflammatory cytokines such as IL-1, IL-8, IL-6, and TNF [71].

The pathogenesis of BD is known to possess an extremely complex genetic background. Apart from the well-known role of the dominant genetic susceptibility factor *HLA-B*51* allele in BD, genome wide association studies (GWAS) and targeted deep re-sequencing of selected immune response-associated genes have allowed the identification of associations between BD and rare variants in *TLR4* (toll-like receptor 4), *NOD2* (nucleotide-binding oligomerization domain-containing protein 2), *MEFV* (MEFV innate immunity regulator), *IL-10*, *ERAP1* (endoplasmic reticulum aminopeptidase1), *STAT4* (signal transducer and activator of transcription 4), *IL-23R* (interleukin-23 receptor), and *IL12RB2* (interleukin-12 receptor subunit beta 2) genes [71,72]. Overall, these findings strongly corroborate the hypothesis that defects in the mechanisms related to the recognition and processing of DAMPs and PAMPs might explain, at least partially, the elevated inflammatory response observed in BD cases.

In recent years, great attention has been given to the characterization of the potential association between Notch signaling and BD. An interesting study conducted by Jian Qi et al. [73] described the findings obtained by deep analysis of Notch signaling in patients with and without active uveitis; this clinical manifestation has been chosen considering that uveitis is typical of individuals with BD. The authors focused on the inflammatory mechanisms and observed a triggering of Notch pathway, with consequent boosting of Th17 response in patients suffering with active uveitis [73]. Furthermore, Jian Qi et al. [73] also observed that the administration of inhibitors of the ϒ-secretase complex were able to significantly reduce the expression of key mediators involved in the inflammatory response (IL-17 and INF-ϒ driven) and in the differentiation of naive CD4+ T-cells into Th17 or regulatory T-cells (STAT3 phosphorylation driven) seen in BD phenotype [73].

### 2.3. Giant Cell Arteritis

Giant cell arteritis (GCA) is one of the major antigen-driven medium and large vessel vasculitis conditions, characterized by episodes of primary vasculitis that lead to the onset of granulomatous inflammatory events principally in the aorta and in its major branches, specifically in the extracranial branches of the aorta together with the blood vessels of the brain, namely spares intracranial vessels [74]. GCA occurs in elder individuals [74] and the susceptibility of the disease has been registered in populations of Northern European descent with an estimated incidence of 15–25 cases per 100,000 people over 50 years of age per year [75]. 

The pathogenesis of GCA is complex and comprehends a strict interplay between genetic and environmental factors; this interaction leads to autoinflammatory episodes in large arteries commonly associated with severe life-threatening outcomes due to the occurrence of events regarding vessel wall remodelling and destruction ultimately manifesting in aortitis and vessel wall inflammation with the concomitant activation of inflammatory-associated pathways [76]. Several genetic studies conducted in the past decade, highlighted that the genetic background in GCA cases was strongly associated with the diseases’ susceptibility and severity [77]. GCA patients frequently possess genetic variants in genes located in the MHC locus and consistent associations have been recognized in MHC-II molecules, specifically in the *HLA-DRB1*04* allele [78]. Further corroborating the strong autoinflammatory response observed in GCA are mutations identified in key mediators of the inflammatory response including *TNF*, *INF-ϒ*, *IL-10*, *IL-4*, *IL-6*, and *IL-18* genes [78]. CD4+ T lymphocytes together with dendritic cells and macrophages seem to be the principal cellular actors involved in the pathogenesis of GCA [76,79]. Specifically, active and mature dendritic cells (DCs) that are retained in the inflamed artery are involved in the production of proinflammatory cytokines, in particular IL-6 and IL-18, and express CD86, co-receptor that allows the interaction and the sustained activation of T-cells in this district [80,81]. Once in the adventitia, resident CD4+ T lymphocytes further sustain the inflammatory response by producing IFN-*ϒ*. The proinflammatory signals induced by the production of IFN-*ϒ* are primarily mediated by the activation of macrophages that, in turn, progressively acquire damaging functionalities including the further production of proinflammatory cytokines, the induction of oxidative damage, nitration of proteins in endothelial cells, and the generation of metalloproteinases that gradually degrade the elastic membranes of vessels [80,82]. 

Recently, tight associations between GCA and the Notch signaling pathway have been observed by Watanabe and collaborators [76]. Upon Notch receptor activation following interaction with Jagged or Delta ligands on a juxtapose cells, NICD translocates to the nucleus and acts as a transcriptional co-activator, thus, regulating the gene expression of downstream target genes [27,83]. In this context, specific crosstalks between mTOR signaling and the Notch pathway seem to provide a novel pathogenic scenario. Indeed, mTOR signaling is specifically involved in strictly coordinating cell growth and metabolism and is rigidly regulated by inputs including cellular energy, nutrients, and growth factors but also by upstream signaling pathways such as Notch signaling [84,85]. Specifically, in GCA an aberrant expression and activation of Notch1 receptor has been registered in CD4+ T lymphocytes and results in an augmentation in the levels of Hes1, a target gene of Notch signaling that functions as a transcriptional repressor [86]. Furthermore, the ligand of Notch1 receptor, Jagged1, has been seen as highly expressed in endothelial cells, induction that is presumably attributed to high circulating levels of vascular endothelial growth factor (VEGF), and therefore strongly contributing to Notch1 mediated signaling pathway activation [76,86]. Currently, a novel hypothesis asserts that the activation of Notch1-Jagged1 signaling axis might trigger a hyperactivation of mTOR signaling activity, ultimately resulting in Th1 and Th17 differentiation and recruitment of T lymphocytes in lesional areas, acting as direct modulators of the disease [76]. The identification of the Notch signaling pathway as a potential chief motif in GCA, might provide a novel target for immunomodulatory approaches also for other medium and large vessel vasculitis.

## 3. Conclusions

To date, only three AIDs characterized by an involvement of Notch pathway have been reported, i.e., hidradenitis suppurativa (HS), Behçet disease (BD) and giant cell arteritis (GCA).

It is widely accepted that the etiopathogenesis of HS is characterized by a deficient Notch signaling; in fact, mutations in the ϒ-secretase complex have a negative impact on Notch signaling [51]. Instead, the etiopathogenesis of BD is distinguished by the triggering of Notch pathway that significantly increases the expression of key mediators involved in the inflammatory response [73]. In addition, Jian Qi et al. also observed that the administration of inhibitors of the ϒ-secretase complex are able to significantly reduce the expression of key mediators involved in the inflammatory response [73]. Additionally, in GCA, there is an aberrant expression and activation of Notch pathway that might trigger a hyperactivation of mTOR signaling activity ultimately resulting in Th1 and Th17 differentiation and recruitment of T lymphocytes in lesional areas [76] (Figure 3).

Therefore, it is clear that a different and somehow conflicting role played by Notch signaling in the inflammatory responses has been observed in the described AIDs. The understanding of the precise roles exerted by Notch signaling in AIDs is important to better understand the different etiopathogenesis of these diseases.

As reported and discussed above, HS, BD, and GCA are all characterized by an involvement of Notch signaling. However, differences in the regulatory mechanisms of Notch cascade have been observed in the three diseases, and thus we cannot identify a common Notch regulatory pathway shared by the three disorders. These differences give rise to several inquiries. The first question is relative to Notch signaling regulation and whether it depends upon distinct factors (possibly under genetic control) influencing Notch pathway in a specific manner for each disease. The second question regards the possibility that, although Notch signaling plays a central but not common role in HS, BD, and CGA, other factors relying on mechanisms other than Notch pathway could be specifically responsible for the pathological different phenotypes characterizing the three AIDs. 

Aimed at clarifying the above-mentioned issues, multi-OMICS studies on large groups of clinically well characterized patients are envisaged. Moreover, the integration between clinical, lifestyle, behavioural, and OMICs findings, including the microbiome and exposome, could provide a well-defined answer regarding the mechanisms involved in HS, BD, and GCA, and thus provide new clues for the design of tailored treatments for patients.

## Figures and Tables

**Figure 1 ijms-21-08847-f001:**
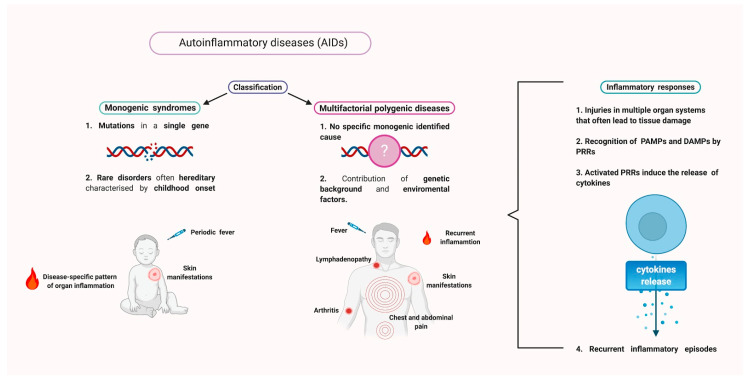
Autoinflammatory diseases. Autoinflammatory diseases (AIDs) are classified in monogenic syndromes and multifactorial polygenic disorders. Monogenic syndromes are often hereditary and rare disorders that encompass mutations in a single gene, and are characterized by periodic fever episodes, skin manifestation, and disease-specific patterns of organ inflammation. Multifactorial polygenic disorders do not possess a specific monogenic identified cause and both the genetic background and environmental factors seem to play a pivotal role in the diseases’ onset. Clinical manifestations of multifactorial polygenic disease include fever, lymphadenopathy, recurrent inflammation episodes, skin manifestations, and chest and abdominal pain. AIDs present injuries in multiple organ systems that frequently lead to tissue damage. In this context, danger signals such as microbial pathogen-associated molecular patterns (PAMPs) and non-microbial damage-associated molecular patterns (DAMPs) are recognized by pattern recognition receptors (PRRs). The activation of PRRs induces the release of proinflammatory cytokines, mainly in innate immune cells, therefore, promoting the development of inflammatory responses.

**Figure 2 ijms-21-08847-f002:**
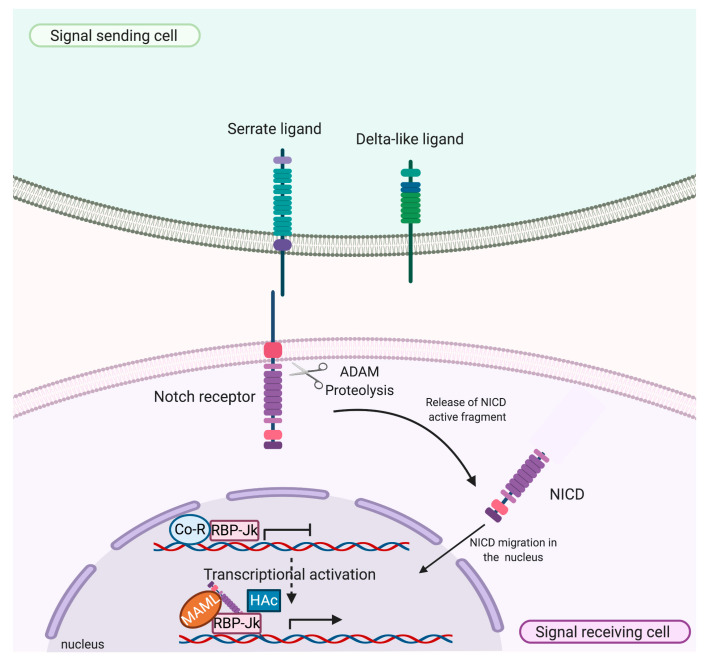
Notch signaling. Notch receptor binds to a Delta-like or Serrate family of ligands located on a juxtaposed cell. Upon ligand binding, Notch receptor is proteolytically processed by the metalloprotease a disintegrin and metalloprotease (ADAM)-type, cleavage that results in the release of the intracellular active fragment, namely the NICD, from the extracellular domain bound to the ligand. Once released in the cytoplasm, NICD translocates to the nucleus and binds to the conserved DNA binding transcription factor RBP-Jκ (recombination signal binding protein for immunoglobulin kappa J region). In absence of interactions with the NICD, RBP-Jκ represses transcription by interacting with the co-repressor (Co-R) complex. Otherwise, transcriptional activation occurs when the NICD binds to RBP-Jκ in the nucleus, therefore, converting RBP-Jκ from a repressor to an activator of transcription through the recruitment of histone acetyltransferases (HAc) and activation of the Mastermind-like (MAML) family, thus, inducing the transcription of downstream primary target genes.

**Figure 3 ijms-21-08847-f003:**
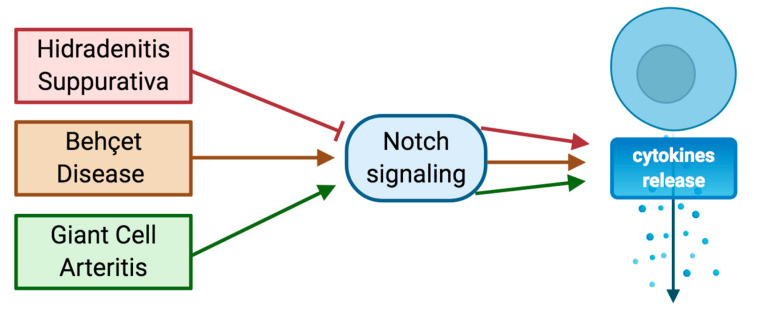
Schematic representation of interactions between autoinflammatory diseases (hidradenitis suppurativa (HS), Behçet’s disease (BD), and giant cell arteritis (GCA)), Notch signaling and inflammation.

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
