# Peer review of "Notch Signaling Regulation in Autoinflammatory Diseases"

_ijms, 2020, doi:10.3390/ijms21228847_

Round 1
Reviewer 1 Report
Gratton et al submit an review article entitled "Notch signaling regulation in autoinflammatory diseases". In this interesting review, they discuss the highly conserved intracellular signalling Notch pathway in the context of three autoinflammatory disorders, Hidradenitis Suppurativa (HS); Behçet Disease (BD); Giant Cell Arteritis (GCA).
Major
the references are presented in an odd manner, with doi and so forth included in the manuscript. This has to be uttterly corrected. Also sse line 106 with "to the development of the disease (Turnpenny)." line 363 Carmona...
In discussion, it would be nice to do a parrallel between notch signalling, NFKB (that is already done by authors) but also ubiquitination and protein degradation, a regulation central to both pathways and important in inflammation and rare diseases. See for example Moretti et al doi: 10.3390/ijms14036359 and Brigant el al doi: 10.3390/ijms20010067 for discussion elements
Minor
line 146 correct please reside not resides
line 337 an augmented triggering is an oxymoron
the english language is somewhat odd and should be correctec. I include a non exhaustive list of awkward sentences.
line 270 In spite of being widely accepted that a deficient Notch signaling characterizes HS etiopathogenesis
line 392 that functions a transcriptional receptor
line 421 It is therefore clear the different and somehow conflicting role-played by Notch signalling in the inflammatory responses of the described AIDs.
line 431 in spite of Notch signaling pathway playing a central, although not common role,
line 433 on mechanisms either than Notch could be specifically responsible
Author Response
We would like to sincerely thank the editor and the reviewers; their criticisms and suggestions really helped us to render the text more readable and to ameliorate the quality of our article, which has been modified accordingly.
Please find included in the text a point-to-point answer to the criticisms raised by the reviewers. Based on your relevant observations we decided to eliminate Figure 2 from the final version of our manuscript, as we believe that this image does not bring any added value to the text.
Reviewer 1
Major
the references are presented in an odd manner, with doi and so forth included in the manuscript. This has to be utterly corrected. Also see line 106 with "to the development of the disease (Turnpenny)." line 363 Carmona.
Response: We apologize for the mistakes, now the references throughout the text have been amended accordingly with journal guidelines for the bibliography. Corrections at lines 106 and 363 have been effectuated as suggested.
In discussion, it would be nice to do a parallel between notch signalling, NFKB (that is already done by authors) but also ubiquitination and protein degradation, a regulation central to both pathways and important in inflammation and rare diseases. See for example Moretti et al doi: 10.3390/ijms14036359 and Brigant el al doi: 10.3390/ijms20010067 for discussion elements
Response: We thank the reviewer for the very interesting observation and for the relevant papers suggested, which we decided to add to the review bibliography. Indeed, the study of the interplay between NF-kB signaling and Notch pathway can contribute to the understanding of inflammatory phenotypes and rare disease. Nevertheless, to date, this axis has not been investigated and characterised in the three disorders reported in our manuscript, therefore, we were not able to discuss further this topic in the diseases’ sections.
Minor
line 146 correct please reside not resides
Response: The text has been accordingly modified.
line 337 an augmented triggering is an oxymoron
Response: The text has been modified as requested.
the english language is somewhat odd and should be correctec. I include a non exhaustive list of awkward sentences.
Response: Following the reviewer’s suggestion, the manuscript has been revised by a native English speaker. We amended all English edits or mistakes.
line 270 In spite of being widely accepted that a deficient Notch signaling characterizes HS etiopathogenesis
Response: The text has been accordingly modified.
line 392 that functions a transcriptional receptor
Response: The text has been accordingly amended.
line 421 It is therefore clear the different and somehow conflicting role-played by Notch signaling in the inflammatory responses of the described AIDs.
Response: The text has been modified as suggested.
line 431 in spite of Notch signaling pathway playing a central, although not common role,
Response: The text has been accordingly adjusted.
line 433 on mechanisms either than Notch could be specifically responsible
Response: The text has been changed as suggested.
Reviewer 2 Report
This is a well-written and timely review, addressing the emerging connection between human autoimmune disease and the Notch signal transduction pathway. Because of the prevalence of AID and the extensive research on the Notch pathway, this piece should be broad interest. I only have some minor comments that may help improve this piece.
I am not an expert on AID, more so on the Notch pathway, so this question may be naïve. But is AID generally thought to result only from immune cell dysfunction, or can other cell types “misbehave” and trigger an otherwise normal immune system to react aberrantly? In other words, when discussing aberrant Notch signalling, in which cells would this occur? Because of the recent emergence of the AID-Notch interplay I would guess that this is not known. But it may help the reader if this was explicitly stated: Where does Notch dysfunction in AID?
This review is generally well-structured and well-written. However, there are some grammatical errors throughout, in particular regarding missing definitive articles (the), and mistakes in using “that” versus “which”.
The text is broken into numerous short paragraphs, and the sub-division is not apparently clear e.g., rows 229-231, making it difficult to follow the flow of the narrative. Obviously, the subdivision of any text into paragraphs is content-dependent, but a rule of thumb is to have at least three sentences per paragraph.
Why both reference [] and DOI, as well as author name sometimes? I guess this will be corrected before final publication, but it made reading the piece cumbersome.
Row 106: What do they refer to with “(Turnpenny)”? Turnpenny-Fry syndrome? Or a missing reference?
Rows 162-170: The Notch pathway is a lot more complicated than outlined herein. For instance, they are omitting the issue of DSL endocytosis, with the NECD domain attached. But maybe this level of detail regarding Notch signalling is appropriate for this type of disease-oriented review.
Figure 3: For clarity, I suggest moving MAML and HAc off of the DNA, since it is generally believed that only RBP/Jk binds DNA.
Rows 244-247: Rephrase sentence.
Rows 241-252: Although the identification of mutations in genes in the gamma-secretase complex in HS patients is very interesting, and indeed implicates Notch signalling, it may be worth pointing out that gamma-secretase also cleaves other membrane proteins e.g., Amyloid Precursor Protein.
Rows 382-383: “juxtaposed”; “to the nucleus”
Author Response
We would like to sincerely thank the editor and the reviewers; their criticisms and suggestions really helped us to render the text more readable and to ameliorate the quality of our article, which has been modified accordingly.
Please find included in the text a point-to-point answer to the criticisms raised by the reviewers. Based on your relevant observations we decided to eliminate Figure 2 from the final version of our manuscript, as we believe that this image does not bring any added value to the text.
Reviewer 2
I am not an expert on AID, more so on the Notch pathway, so this question may be naïve. But is AID generally thought to result only from immune cell dysfunction, or can other cell types “misbehave” and trigger an otherwise normal immune system to react aberrantly? In other words, when discussing aberrant Notch signalling, in which cells would this occur? Because of the recent emergence of the AID-Notch interplay I would guess that this is not known. But it may help the reader if this was explicitly stated: Where does Notch dysfunction in AID?
Response: In general, AIDs are defined as disorders characterised by an aberrant and deregulated innate immune response without an involvement of adaptive immunity (auto-reactive T-cells and auto-antibodies), presenting spontaneous sterile inflammation in multiple organ systems [1]. In the context of AIDs, due the extremely vast heterogeneity of these disorders, different possible scenarios might occur. The presence of mutation in both inflammation-related and / or alternative genes might occur in innate immune cells or in other cell types that might trigger, through various disease and/or site-dependent mechanisms, the innate immune system to react in a deregulated manner.
Specifically, while discussing of aberrant Notch signalling the type of cells in which this deregulation occurs is strictly disease-dependent and in most cases still not known. Therefore, based on current knowledge it is not possible to universally and uniquely state.
[1] Ben-Chetrit, E.; Gattorno, M.; Gul, A.; Kastner, D.L.; Lachmann, H.J.; Touitou, I.; Ruperto, N. Consensus proposal for taxonomy and definition of the autoinflammatory diseases (AIDs): a Delphi study. Annals of the Rheumatic Diseases 2018, 77, 1558–1565, doi:10.1136/annrheumdis-2017-212515.
This review is generally well-structured and well-written. However, there are some grammatical errors throughout, in particular regarding missing definitive articles (the), and mistakes in using “that” versus “which”.
Response: Following the reviewer’s suggestion, the manuscript has been revised by a native English speaker. We amended all English edits or mistakes.
The text is broken into numerous short paragraphs, and the sub-division is not apparently clear e.g., rows 229-231, making it difficult to follow the flow of the narrative. Obviously, the subdivision of any text into paragraphs is content-dependent, but a rule of thumb is to have at least three sentences per paragraph.
Response: We have modified the text in order to ameliorate the flow of the narrative by joining shorter paragraphs that are coherent with the further described section.
Why both reference [] and DOI, as well as author name sometimes? I guess this will be corrected before final publication, but it made reading the piece cumbersome.
Response: References have been amended throughout the text.
Row 106: What do they refer to with “(Turnpenny)”? Turnpenny-Fry syndrome? Or a missing reference?
Response: Turpenny refers to a missing reference. The text has been appropriately modified.
Rows 162-170: The Notch pathway is a lot more complicated than outlined herein. For instance, they are omitting the issue of DSL endocytosis, with the NECD domain attached. But maybe this level of detail regarding Notch signalling is appropriate for this type of disease-oriented review.
Response: We agree with the fact that the way that Notch pathway was presented in the review does not reflect the true level of complexity of the signalling route. This is an aspect we have deliberately decided to assess during the redaction of the manuscript since we are convinced that further levels of detail are not required for this type of disease-oriented review.
Figure 3: For clarity, I suggest moving MAML and HAc off of the DNA, since it is generally believed that only RBP/Jk binds DNA.
Response: The figure has been accordingly modified.
Rows 244-247: Rephrase sentence.
Response: The sentence has been rephrased as suggested.
Rows 241-252: Although the identification of mutations in genes in the gamma-secretase complex in HS patients is very interesting, and indeed implicates Notch signalling, it may be worth pointing out that gamma-secretase also cleaves other membrane proteins e.g., Amyloid Precursor Protein.
Response: The text has been implemented with the suggested information at lines 328-329.
Rows 382-383: “juxtaposed”; “to the nucleus”
Response: The text has been amended as suggested.